# Developing Computational Thinking Teaching Strategies to Model Pandemics and Containment Measures

**DOI:** 10.3390/ijerph182312520

**Published:** 2021-11-28

**Authors:** Roberto Araya, Masami Isoda, Johan van der Molen Moris

**Affiliations:** 1Center for Advanced Research in Education, Institute of Education, Universidad de Chile, Santiago 8320000, Chile; 2Faculty of Humanities and Social Sciences, University of Tsukuba, Tsukuba 305-8577, Japan; isoda@criced.tsukuba.ac.jp; 3MRC Biostatistics Unit, University of Cambridge, Cambridge CB2 0SR, UK

**Keywords:** COVID-19, computational thinking, computational modeling, lesson study

## Abstract

COVID-19 has been extremely difficult to control. The lack of understanding of key aspects of pandemics has affected virus transmission. On the other hand, there is a demand to incorporate computational thinking (CT) in the curricula with applications in STEM. However, there are still no exemplars in the curriculum that apply CT to real-world problems such as controlling a pandemic or other similar global crises. In this paper, we fill this gap by proposing exemplars of CT for modeling the pandemic. We designed exemplars following the three pillars of the framework for CT from the Inclusive Mathematics for Sustainability in a Digital Economy (InMside) project by Asia-Pacific Economic Cooperation (APEC): algorithmic thinking, computational modeling, and machine learning. For each pillar, we designed a progressive sequence of activities that covers from elementary to high school. In an experimental study with elementary and middle school students from 2 schools of high vulnerability, we found that the computational modeling exemplar can be implemented by teachers and correctly understood by students. We conclude that it is feasible to introduce the exemplars at all grade levels and that this is a powerful example of Science Technology, Engineering, and Mathematics (STEM) integration that helps reflect and tackle real-world and challenging public health problems of great impact for students and their families.

## 1. Introduction

The ongoing COVID-19 pandemic has demonstrated that most of the knowledge regarding the growing whirlwind of scientific and technological advancements has not been adopted by the general population. Despite great efforts to stop the transmission of the virus, we have seen that in the majority of countries all over the world, this has not been easy [1,2]. Certainly, this is a multifaceted problem. Nonetheless, education is clearly one of the main aspects that can help explain the behavior and attitudes of the population in this context. Therefore, there is now an increasing demand to educate students to understand pandemics and help students to change some of their behaviors and attitudes [3]. It is then important to find ways to design educational strategies to prepare society for this and other global crises.

However, we have to consider that nowadays we are also experiencing several other demands to introduce changes in the science and mathematics curricula. One of the new demands is the introduction of computational thinking (CT) in K12. CT is a problem-solving process that includes (but is not limited to) identifying patterns, logically organizing and analyzing data, making abstractions, developing algorithms, and formulating problems for computers to be able to solve them [4]. Currently, there is a curriculum reform movement in this era, of artificial intelligence (AI) and Big Data for the 4th industrial revolution [5]. CT is becoming a key factor across the curricula in order to establish a digital economy [6]. CT is increasingly considered a critical cognitive disposition. Indeed, it is critical to solve the challenges of the digital society. However, CT requires some basic level of literacy on how computers will process the information, how they connect with the internet of things (IoT) and smartphones, and how they integrate AI and Big Data. CT is closely related to computer and mathematical sciences. In a digital society, it is necessary that citizens have the skills and abilities to observe and assimilate various real-world phenomena as information and make connections appropriately and effectively to discover and solve problems, forming their own ideas. Mathematics functions as a language and provides ideas to use information in order to achieve this [7,8]. In this context, mathematics is recognized as a key subject to develop computational thinking. Thus, it is necessary to reorganize and adapt the mathematics curriculum to the current era. Therefore, we need to specify how CT can be adopted in the required educational reform; and, to what extent it can be introduced at different school levels in the curriculum.

Even though we are immersed in a digital society with mobile apps that constantly accompany and help us, one of the challenges when introducing CT is to connect it with the rest of the school curriculum [9,10]. Another challenge is to connect it with the main issues that motivate schoolchildren [11,12]. The pandemic greatly affects the life and well-being of each one of us, thus, it offers a solution to the key problem of motivation and meaning, since in general it is difficult to find a problem that is most affecting the daily lives of students.

Another curricular demand is more integration across disciplines. This demand originates from the greater integration in the same disciplines. For example, the essence of the new biology is the integration of several disciplines; not only between its many sub-disciplines but also with physics, chemistry, computer science, engineering, and mathematics. One key strategy is to emphasize the use of cross-cutting concepts [13] such as computation, information, patterns, and mathematical and computational models. Additionally, visualization is also key in the integration of disciplines [14]. This is called STEM education since it integrates Science Technology, Engineering, and Mathematics. If Art is included, it is called STEAM. It enhances design for creativity in addition to logical and critical thinking. For example, simplicity in new designs has become a keyword for computer-based modeling and visualization [15]. The pandemic also offers a great opportunity for the integration of disciplines, ranging from biology, mathematics, computer science to social sciences.

However, to take advantage of these opportunities, it is necessary to propose precise exemplars. They are more than just examples. They are outstanding and exceptional examples of lessons that illustrate very well a central concept from one of the pillars in a curriculum. They should be activities that can be used at different levels (from elementary, to middle and high school), and should ideally be a sequence of increasing depth as the student moves from lower levels to higher levels. Additionally, it is crucial that they are activities aligned to the math and science curriculum. Exemplars should offer an opportunity to reflect and understand more deeply the core math and science concepts of the curriculum.

These three curricular demands, CT, STEM integration, and pandemics, can now be synergistically integrated since several computational models are constantly being published in the mass media and written press with graphs and dynamic simulations under different conditions. They are constantly discussed by experts in terms of their predictions and practical implications. Thus, citizens are aware that computational models are key tools that help define health and economic policies [16]. Concepts such as “flatten the curve”, “social distance”, and “face covering” have become part of the everyday language, as a consequence of the implementation of these policies. On the other hand, it is unclear to what extent the general public understands the connections between individual actions including the use of face masks or social distancing, and the global aspects of the disease transmission on which computational models are based. These global or aggregated aspects of the pandemic are usually represented by abstract terms including “curve” and “parameter”.

Furthermore, people always have mental models of a phenomenon [17], even if they are not aware of them. They are implicit models, and what the citizens do depends largely on those models. Measures are necessary, but they are not sufficient. For instance, some people might have a mental model of transmission such that it only occurs if there is an intention of one subject to infect another. In this way, if we belong to a community of well-known and well-intentioned people, then even if we meet in a small place and without masks, we should not be infected. The government can deliver very effective measures, but if the infection model the people have includes their own beliefs, then people will not follow those measures. Two examples of this are the causation model of leprosy in Europe in the twelfth century [18], and the Nigerian case of impact of witchcraft belief of disease causation [19]. Health authority measures based on correct models are also critical. One example is the case of pellagra, a disease that spread across Europe when the poor population relied on a mainly corn diet after discovering and bringing corn from the new world. Europeans thought it was spoiled or moldy corn and developed a wrong causal model [20]. They even passed laws that prohibited the sale of spoiled and moldy corn, but these official measures did not work. Scientific reasoning is a stronger independent predictor of unfounded beliefs (including anti-vaccination attitudes) than general analytic thinking is [21]. All these examples indicate that citizens having correct mental models of transmission mechanisms has an important role in changing citizens’ behavior and adoption of measures suggested by the health authority. Given the critical role that computational models play in the definition of public health and economic policies, the effectiveness of concrete measures derived from such policies relies on a good understanding of the connection between these abstract models and practical individual actions.

Moreover, these models are usually adjusted and updated using real data collected from different sources. Thus, they are also linked to the ability to make decisions based on evidence provided by these data sets, and to critical thinking to evaluate the validity of the data and the model outcomes. Additionally, it is necessary that the means by how information is communicated are well understood; graphs and methods of data visualization, in general, are therefore another subject that has a direct impact on people’s decisions.

Hence, a good educational strategy to tackle future global crises should encourage students to develop skills necessary to understand at least the core ideas, assumptions, simplifications, biases, limitations, missing components, and simulation strategies of the computational and mathematical models of disease transmission by a viral infection, as well as the corresponding data visualization and the appropriate evidence-based decisions to control it.

We have to consider that the change required is not just one of knowledge and skills. It is also critical to achieve a change in the values and attitudes of the students. Values and attitudes are also critical components of curricula worldwide [22]. The current era is considered as the era of globalization under new societal values on sustainable development goals (SDGs) by the United Nations Educational, Scientific and Cultural Organization [23], and by other organizations such as the Southeast Asia Ministers of Education Organization [24]. These values are in the process of being embedded into school textbooks all over the world [25]. The problems of COVID-19 are directly related to SDG No.3 and partially related to other SDGs. Global citizenship education is an issue across different SDGs. These issues and the matter of citizenship have been enhanced as critical in mathematics education and pedagogy [26,27,28,29]. However, it is difficult to measure attitudes and values [30]. Nonetheless, teaching global world issues with learning activities, class discussions, and time dedicated to those problems helps students change values and attitudes [31]. Thus, our research question is: Is it feasible to implement a computational thinking teaching strategy to model pandemics, and to what extent do students understand and reflect on the factors that influence pandemics and ways to control them?

To answer these questions, there are several approaches. The first approach is to clarify the current meaning of mathematical sciences in relation to the adaptation of mathematics to the computer sciences. Indeed, it is an ongoing effort at the higher education reform on mathematics education [32,33]. The second approach is to review mathematical thinking from the perspectives of computational thinking. Traditional mathematical thinking has been considered in the context directed to pure mathematics [34,35] and in the current curriculum these issues are already considered and some revisions have been made. For example, they are in the process of reconsideration in Southeast Asia [36]. The third approach is considering embedding informatics for AI and Big Data into the mathematics curriculum. APEC InMside project [37,38,39,40] for regional curriculum reform focused on this third approach in relation to CT [41,42] and statistical thinking [43].

From the three approaches described above, the first one only applies to higher education, the second is still being studied and under revision, therefore we chose the third one as it is suggested by APEC and it is the most suitable for our context. Indeed, in the InMside project (2019), curriculum specialists for informatics and mathematics in the APEC region produced new perspectives for curriculum reform on discussion documents for computational thinking [41] and statistical thinking [43] and produced the report for curriculum reform on the APEC region [44].

The plan is to help design lessons to introduce computational thinking in the mathematics curriculum as part of the design of educational strategies to prepare society for global crises, and in particular for pandemics. Therefore, our methodology is to apply the APEC InMside framework to create new exemplars of CT-based learning activities consisting in solving problems related to the pandemic and adapted to students of elementary, middle, and high school.

In the following sections, we describe the framework used, show the exemplars explaining their corresponding CT and public health-related concepts, report the results on an experimental implementation in two schools using the lesson study methodology [45], and finish with our conclusions.

The main contributions of this paper are:It is feasible to integrate computational thinking teaching with real-world problems such as the COVID-19 pandemic (and its corresponding educational challenges).We provide empirical evidence to support that○Students are able to analyze a computational model and connect with the corresponding graphs of the progression of the transmission phenomena.○Students can understand how the transmission mechanism of the computational model includes critical factors such as population density, population mobility, facemask use, and social distance.○Students can be aware of some of the limitations of the computational model. They can reflect on these limitations and devise ways to overcome them.

## 2. Framework: The Three Pillars of Computational Thinking

The APEC InMside project [40] proposes a framework with three pillars for CT, developed in collaboration with curriculum specialists from the APEC region [42]. In this framework, CT is defined as an approach to problem solving that consists of analyzing, simplifying, and automatizing the solving process. The framework considers three pillars: algorithmic thinking, computational modeling, and machine learning (see Figure 1).

The first pillar is algorithmic thinking, which means decomposing a problem into very simple instructions that can be followed unequivocally by any person or a computer. One of the key ideas for algorithmic thinking is recursive reasoning [46,47]. Algorithmic thinking comes from detecting repeated patterns in problems. Then, we describe those patterns as a set of rules for dealing with such situations (so that one does not need to think this through each time the problem occurs). Effectively, one creates (formally or informally) an algorithm or rational procedure. This kind of thinking is often used in problem solving and in computer programming. Especially, if algorithmic thinking is represented by a program, there is a large number of mathematical techniques to be recognized as an algorithm [48,49,50]. Indeed, an algorithm is a list of simple instructions that anybody can understand and even a machine can follow [51]. Typically this can be carried out with a pseudocode, logical operators, lists and arrays, and cycles. Infinite loops are particularly powerful since the machine can run them without getting tired or bored [52]. Moreover debugging, testing, documentation, peer review, and teamwork are also essential for programing.

The second pillar of the framework comes from science where CT is regarded as building computational models [53]. Here, “computational” means “programmable” which includes “mathematical”. This type of CT is behind the current progress in biology, where increasingly accurate computational models of cells and organisms are being built. This is closely related to COVID-19 in various ways since computational and mathematical models are used to study the dynamics of viral infection with the aim of predicting the number of infected and recovered people through time, as well as the total number of deaths. Moreover, swift development of a vaccine against the virus SARS-CoV-2 has been possible by using computational models that help to predict which parts of the virus would elicit an immune response. According to John Holland [54], a pioneer in the application of cellular automata to biology and the creator of genetic algorithms, patterns are normally expressed with board games. He claims that board games are as equally important to the scientific endeavor as numbers. He states that board games, as well as numbers, mark a watershed of human perception of the world. Holland also introduced computational models that can evolve, such as in natural selection, and thus are able to solve problems that their creators do not fully understand. Students can start with a concrete cardboard model, with a board, cards, chips, turn-taking rules, and simple rules that characterize a phenomenon. It is also necessary to design ways to test that the model does what is required to do, compare its predictions with the natural phenomena, debug it, document it, and share tasks with a team.

The third pillar comes from data science and machine learning (ML), which is at the core of the 4th industrial revolution, which will have a significant impact on the nature of jobs and employment. It represents a change of paradigm as well. Users train the computer instead of programming it. According to the Fields Medal mathematician, Cedric Villani, ML techniques lead the gradual transition from a programming logic to a learning logic [55]. ML is at the core of AI. Most of what is labeled AI today is actually ML [56]. This is an algorithmic field that blends ideas from statistics, computer science, and many other disciplines. Core ideas are understanding and recognizing in a given problem what are the relevant features, what are the classes or categories to be identified or detected, metrics to assess the discriminating power of features or combinations of them, visual representations such as scatter graphs, construction versus test databases, and measurements of the performance of the trained classifiers. Skovsmose [26,27] highlights cross-cutting concepts in critical mathematics education, such as abstraction, modeling, algorithm, and technological imagination. However, ML provides a change in the logic from programming to learning which will be a new aspect for critical mathematics education with applications.

Figure 1 depicts the curriculum framework to develop CT through a conceptual diagram. This development is supported by the three pillars shown as the sides of the central triangle. On top of each pillar, there are three keywords that represent them. CT can be developed in activities from every subject in the curriculum and also crosscutting many of them.

## 3. Materials and Methods

We propose the lesson study methodology. This methodology began in 1880 in Japan with the aim of reproducing best practices in teaching [45]. It has been adopted in several other countries [57,58]. It is a collaborative teaching improvement process designed to generate strong and productive communities of teachers sharing and learning from each other (Figure 2).

Thus, we designed exemplars of lessons for each of the three pillars of CT. Then, using the methodology of lesson study, we implemented one exemplar in several elementary and middle school classes. We conducted it in two high-risk schools with a high level of vulnerability. A team of four elementary and middle school teachers worked on the lesson study implementation. The teachers, together with us, held constant meetings to review the sessions to reflect and adjust the lessons.

The exemplars proposed are learning activities aimed at students of different school levels. They correspond to solving problems related to the current COVID-19 pandemic using CT. We provided not only examples of exercises and activities but also guidelines to put them into practice in the classroom. We show an exemplar for each pillar of the framework presented in the previous section, with variants adapted to three school levels, elementary, middle, and high school. In each case, we highlight the relevant CT concepts and their connection to critical mathematics education and curriculum, as well as the connections to significant practical public health elements.

### 3.1. Exemplar 1: Algorithmic Thinking to Estimate the Number of Infected People

One of the main problems in the current pandemic is to know the number of infected people (including asymptomatic cases) in each country, every day. The typical data published in the press and web pages [59,60] show cases that are PCR tested, PCR-positives cases, and deaths. However, it is almost impossible to find information on the number of infected people. This information is critical to estimate the level of spread of the virus, and the trends on its transmission to determine if the corresponding curve is flattening. The number of tested individuals and the number of positives are not enough to estimate the percentage of the infected population. Given the fact that the set of tested people are those who already feel sick and therefore suspect of being infected, the set of tested people is biased and it makes it very difficult to estimate the true number of infected people. Furthermore, given that some countries carry out many tests and others very few, it is very difficult to compare the figures of PCR-positive cases between countries and estimate trends based on comparisons across countries.

Thus, the problem has critical significance to the everyday life of every citizen (Ernst, 2010). Avoiding wrong conclusions from this kind of data requires deep critical thinking, together with CT, proportional reasoning, ideas of probability, and statistical inference [61]. Here, we propose three learning activities related to the problem estimating viral transmission and the proportion of infected people in a given population.

#### 3.1.1. Elementary School Lower Grade Students Who Learned Numeration and Counting

In order to control the pandemic, it is critical that students understand the transmission mechanisms. One feature of algorithmic thinking is proceduralization by using recursion. For example, the following elementary school coloring activity contains recursive instructions in order to understand disease transmission. One simple rule—to color in red the balls that touched another red ball (or balls)—has to be recursively applied (Figure 3).

Some relevant questions from this activity are: In which of the two tables there will be more transmission? How many balls will be painted red on each table? In how many moments will the transmission stop at each table?

In order to solve correctly this task, students need to understand the rule, and in particular, associate the concept of touching with having a common point of tangency. They also have to produce a group of elements defined by a certain condition, and therefore reflect on mathematical concepts such as recursion and set [36,62,63]. They also need to identify a stopping condition for the recursion and interpret the number of moments or steps as the number of times the rule is applied recursively until it stops. Students can discuss how they applied the rule and the results they obtained. By explaining their reasoning, they can improve their understanding of the key concepts and help others to correct mistakes.

#### 3.1.2. Middle School Students Who Learned Ratio and Proportion

The problem in the previous activity was to identify who is infected (the red balls). Thus, for middle school students, we propose an approach to estimate the true proportion of infected individuals. It is based on the use of data gathered in particularly selected ideal conditions, where all the people have been polymerase chain reaction (PCR) tested and followed for several weeks. One such case is the Princess Diamond Cruise that was stranded off the coast of Japan for several weeks. Another case was several returning Wuhan residents on repatriation flights that arrived in the city and where all passengers were PCR tested, remained in quarantine, and tracked for several weeks. From all these special cases, it can be estimated [64] that 0.66% of the population with a PCR-positive test will die. This is similar to Gates [2] estimate of 1%. This means that one person dies out of 150 that were positive in the PCR test. Moreover, on average the person dies 14 days after being tested [65].

Let us consider a country where one out of 15 positives in the PCR test dies 14 days after being tested. Here, instead of coloring (Figure 3), the scenario can be represented with a ruler and wooden blocks to facilitate reasoning and help make clear the algorithmic-computational process. The ruler represents days, which starts with day 0 representing the day an individual is PCR tested. Each individual is represented by one block. Then, the described scenario where from 15 PCR-positives one dies 14 days later is represented by stacking 15 blocks on the zero mark of the ruler and placing one block on the mark corresponding to 14. This is shown in Figure 4.

However, this country—as all countries—does not test all people every day. Then, how can we use this information to estimate the number of infected people in the country? This question is represented in Figure 5, where the unknown number of infected at day 0 is represented with a pile of blocks.

How can we estimate the number of infected people on a given day? Given that from 150 people one will die in 14 days (according to the rate obtained from the ideal cases), and that in this country from 15 PCR positives one will die in 14 days, then there is an opportunity to carry out proportional reasoning. First, the key is to focus the attention on the person that dies on the 14th day. Then, think backwards [34] in time, to 14 days earlier. On one hand, there are 150 infected people. One person of these 150 dies 14 days later. Therefore, there are 150 people infected for the one subject that dies and to whom the attention is placed. On the other hand, there is a particular statistic for the country. From 15 PCR positives, one person dies 14 days later. Therefore, there are 15 PCR positives related to the one person that dies 14 days later. This means that there are 150 infected and 15 PCR positives. Hence, for every 10 infected people in the country, there is 1 PCR positive case.

The activity for students consists of filling out a spreadsheet for their country, registering for each day the number of PCR-positive people and the number that die 14 days afterwards. Moreover, from this information, students can deduce the rule for the changing number of infected people on each day in their own country.

This activity also requires algorithmic thinking. Students have to consider and select appropriate data for every day and connect it with data with a delay of 14 days. They also have to gather data and arrange it in tables, search for patterns, combine population and proportional thinking, and input mathematical expressions in such a way that can be automatically updated.

Students can code the algorithm with spreadsheet functions in order to automatize the process. Thus, the students have to translate the mathematical relations in the specific syntax of the spreadsheet language. This translation is an important component of mathematics education [26] present in curricula such as the ASEAN curriculum standards [36].

#### 3.1.3. High School Students Who Learned Conditional Probability

For high school students, we propose a different lesson plan than the previous one. Here, we PCR test a random sample of the population, and for this task, we will explore two approaches, natural frequencies [66] and single-event probabilities. We will work under the frequentist Neyman–Pearson paradigm, in which probabilities are defined as the outcome of the limiting frequency with which that outcome appears in a long series of similar events [67].

Understanding the meaning of probabilities contributes to reducing statistical illiteracy which can have serious consequences for public health [66]. The way in which probabilities and risks are represented and communicated is critical. As emphasized by Cosmides et al. [67] “from an ecological and evolutionary perspective, certain classes of problem-solving mechanisms in the human mind should be expected to represent probabilistic information as frequencies”. The natural frequency approach proposed by Gigerenzer [66] improves statistical literacy [68] not only in frequentist frameworks but also in Bayesian setups [69]. This approach is an ecologically valid format, that uses frequency statements instead of single-event probabilities and it helps to infer correct probabilities in typical tests and diagnosis situations. Evidence shows that subjects that do not actually use natural frequencies for their calculations but instead translate them back into traditional expressions for probabilities, usually produce wrong answers [70].

Two key parameters of any viral test are its sensitivity and specificity [71]. Sensitivity is the percentage of people who are infected who test positive. Specificity is the percentage of people who are not infected and who test non-positive. Figure 6 provides a visualization of a test that has a sensitivity of 70% and specificity of 99%. In this example, there are 100 infected people and 100 non-infected people. All people have been tested and from the 100 of them that were infected 70 of them are PCR positive, and 30 are PCR negative. From the total number of non-infected people, 99 of them are PCR negative, and only one is PCR positive.

Furthermore, assume that in a given country on a given day, 10% of the population is infected. Let us assume also that the population is large enough so that considering individuals independent or sampling them with replacement provides a good approximation of the true sampling process for testing (without replacement). If we select a random sample of 100,000 individuals, then most likely about 10,000 individuals will be infected, and therefore about 7000 of them will most likely be PCR positive, and about 3000 will be PCR negative. Out of the 90,000 non-infected individuals, 89,000 will most likely be PCR negative and 1000 will be PCR positive. This means that in total we will most likely have 8000 PCR positives and 92,000 PCR negatives.

It is interesting to note that 7000 out of 8000 PCR-positive individuals are indeed infected. Moreover, 89,000 out of the 92,000 PCR-negative individuals will be not infected. This means that 3000 out of the 92,000 PCR negatives are infected.

Given that we want to estimate the true proportion of infected people in a given day, we have to consider this number as unknown, whereas the proportion of positives is known. Let us call X the percentage of infected people. This means that X out of every 100 people are infected and let us call P the percentage of PCR positives. This means that P out of every 100 people are PCR positives. Let us also assume that the sensitivity is A, this means that out of 100 infected people A are PCR-positives. Finally, let us assume that the specificity is B, this means that out of 100 not infected, B are PCR negatives.

Following the same reasoning as in the example, if we test a random sample of 100,000 people, we would most likely obtain X × 1000 infected, and from those A/100 × X × 1000 = AX × 10 will be PCR positives (by using the rule of three on two ratios). On the other hand, we would obtain (100 − X) × 1000 non-infected people and from those, ((100 − B)/100)(100 − X) × 1000 = (100 − B)(100 − X) × 10 would be PCR positives. Then, in total we would obtain [AX + (100 − B)(100 − X)] × 10 PCR positives out of the 100,000 tested people.

Thus, the number of PCR positives every 100 people is given by:P = (AX + (100 − B)(100 − X))/100,(1)

Hence, the number of infected in 100 people is:X = (100P + 100B − 10,000)/(A + B − 100) = 100(P + B − 100)/(A + B − 100).(2)

Alternatively, the single event approach expresses the probability of a person being infected as a real number between 0 and 1. We denote this unknown probability as x. Students should also consider another pair of events, obtaining a positive or negative result on the PCR test; and denote the probability of a person testing positive as p. Furthermore, let us call a the sensitivity of the PCR-test which means that a is the probability of being positive given that the person is infected. Additionally, let us call b the specificity of the PCR test which means that the probability of being negative is b given that the person is not infected. Now, using the law of total probability, we obtain:P(positive) = P(positive│infected)P(infected) + P(positive│non-infected)P(non-infected),(3)
where, according to our notation, P(positive) = p, P(infected) = x, P(non-infected) = 1 − x, P(positive│infected) = a, and P(positive│non-infected) = 1 − b. In this way, we obtain the formula:p = ax + (1 − b)(1 − x),(4)
from which we can find x, namely:x = (p + b − 1)/(a + b − 1),(5)
which as expected, is the same formula found for X setting x = X/100, p = P/100, a = A/100, and b = B/(100).

Additionally, students can generate their own simulated country using a spreadsheet. They can create a list of tested people stating whether their result was positive or negative. In order to use the formula, they need to know the sensitivity and specificity of the test for which they can also create two lists, one with 100 infected people and the other with 100 non-infected people and generate their test results. After simulating the proportion of positives in the sample of tested people in the county, and the two parameters of the test, they can apply the formula to find the estimated proportion of infected people in the population.

Finally, students can also plot how the proportion of infected people changes as a function of the proportion of positives for fixed parameters of the test; or plot and analyze the effects of changing either the sensibility or specificity while keeping fixed the other two variables.

### 3.2. Exemplar 2: Computational Modeling on Population Density, Mobility and Wearing Face Masks

Each day, mass media publish predictions on the evolution of the pandemic based on simulations using computational models [72,73]. Citizens hear keywords such as “models”, “computational” and “simulations”, and they are told that these computational models are able to predict the future evolution of the pandemic. However, not all predictions agree since they are based on slightly different assumptions. For example, most of them do not consider the effect of different strategies [74], including intensity of sampling for PCR tests, the closedown of schools, restrictions on the number of people that can attend an event, partial or complete quarantine, reopening [75], occupancy of hospitals, density of the population, restrictions on mobility [76], use of face masks, mutation of the virus, etc.

The goal of this exemplar is to foster students’ reflection on the impact of social distance on the spread of the pandemic and try to understand the mechanisms that generate different propagation rates. The activity is divided into three parts. First, for elementary school, students reflect on possible variables that define social distance and build simple concrete models. Second, middle school students reflect on how to quantitatively estimate the impact of the variables in the propagation of the contagion. Third, high school students reflect on how to translate the concrete physical model into a computer model.

Building models is a basic thinking activity [17]. Most models are implicit, however, in computational modeling, we need to make them explicit, computationally. A model is a simplification of the phenomenon but still complex enough to be able to capture the main characteristics and behaviors [77].

#### 3.2.1. Elementary School Students Who Learned Counting

In order to understand the effects of population density and social distance in the propagation of the virus, a physical model is used to illustrate them. This activity begins with a short video to explain the possibility of getting infected at the supermarket and asks students to explore strategies, setting the rules to decrease transmission. Lower-grade students can model a classroom or supermarket as a rectangular pool table, and people as balls. Upper-grade students can pose questions about population density and its effect on virus transmission. Higher population density is represented with a higher number of balls on the pool table and higher population density represents a higher risk of infection. For example, Figure 7a–c show three scenarios with a different number of balls [78]. Additionally, balls moving on the pool table can function as a metaphor for the movement of students in a classroom or people in a supermarket, and virus transmission is the collision of balls.

Then, students have to predict in which scenario Figure 7a–c there will be more infection or collisions. With a smartphone, they record a 30 s video on each of the three scenarios and then they review the videos and count collisions. At first, only one ball moves in a random direction while all the others remain still. To facilitate this task, students can place a cardboard cone on the table, place the ball on top of it, and release it. This also helps to reproduce the experiment with a fixed speed but setting a different direction each time. In this way, students can reflect on how the population density affects the level of infection. Next, they have to reflect on the impact of higher population mobility; they model it as the higher speed of the balls, by using cones with a bigger height, or more balls moving at the same time, using several cones. By trying different values of one variable while fixing the other, students can conclude that higher mobility implies higher infection levels.

Although this model captures very well the relation between mobility and transmission, it misses other relevant variables. Human behavior depends on cultural practices, thus cultural differences may produce different infection outcomes. For instance, Figure 2 shows how Japanese schools have a custom of using masks in the winter season to avoid lockdowns due to influenza. Isolation strategies and wearing personal protective equipment (PPE), such as wearing face masks, help decrease the propagation of viruses [79]. Now, we present how to include PPE as a variable in our model.

Students model individuals using face masks with smaller balls than the balls representing individuals without masks. This means that in order to have transmission between two people wearing masks they need to be very close to each other, but for two individuals without masks transmission can be produced even when they are farther apart. Different levels of protection can be modeled by different ball sizes. For example, ordinary masks are not very effective against inhaling air with a high viral load, however, it effective in preventing the spread of the virus to larger areas due to sneezing, whereas higher quality masks such as N95 respirators are highly efficient at filtering airborne particles and so they can prevent transmission even at close proximity to an infected person. In our case, ordinary masks will be represented by balls of a larger radius compared to the ones representing higher quality PPE but will be smaller than representing people with no masks at all.

Next, students make predictions and experiment recording on video in a similar way to the previous activity these new conditions, small and larger balls, as shown in Figure 8. Then, they count the number of collisions.

#### 3.2.2. Middle School Students Who Learned Means and Ratios

For these students, the activity consists of repeating first the previous activity and then moving further in the analysis from a physical model to a computational model estimating the collision probabilities in the different conditions of population density and the use of face masks, for a fixed level of mobility. They can represent the social distance mathematically. For example, they can compute the proportion of free space on the table. Small balls produce a larger remaining space and large balls produce smaller remaining space in the pool table. The remaining space can be represented by subtraction or a ratio. Now we show such new quantities. Middle school students use the numerical results and think inductively. They can also discuss how their produced quantities represent a risk of infection.

For simplicity, we set the space of the pool table as 100 m^2^, and denote the area covered by a ball of radius r as x m^2^, and the number of people as *n*. One possible ratio is:(6)$$\frac{Number\;of\;Population}{Area\;\left(pool\;table\right)-Area\;\left(total\;occupied\;space\right)}=\frac{n}{100-nx}=\frac{n}{100-n\pi r^{2}}.$$

Students can compare two situations with different numbers of balls (people) but the same remaining space. They can also discuss in what sense this ratio captures the risk of transmission considering the space that can potentially be occupied by any of the balls, which in turn depends on their size.

Another quantity they can analyze is the ratio of remaining to occupied area:(7)$$\frac{(Area\;\left(pool\;table\right)-Area\;\left(total\;occupied\;space\right)}{Area\;\left(total\;occupied\;space\right)}=\frac{100-nx}{nx}=\frac{100}{nx}-1=\frac{100}{n\pi r^{2}}-1.$$

Again, this odds ratio depends on the number of *n* people and their radius of transmission, and students can reflect on how higher values of this quantity represent lower chances of transmission. Another related problem is to find the maximum number of circles of a certain radius that can fit onto the rectangular space.

They can compare the previous ratio with its reciprocal that corresponds to the odds ratio of occupied space over free space.

Next, students study the relation between the speed of movement and risk of infection. They choose one of the balls at rest and place a strip of paper representing its movement trajectory. The strip of paper is a rectangle with one side equal to the width of the ball (that is, its diameter), and the other side equal to the length of the path traveled. Then, students count the number of collisions, that is, the number of balls at rest that are in that trajectory, and repeat this with strips of different lengths. Finally, students can also calculate the area occupied by the strip of paper, and calculate the ratio between the occupied and total area of the table. They then reflect on whether that ratio represents the probability of a collision with the remaining balls at rest.

Additionally, students can explore the effects of other ways of increasing mobility, modeling multiple balls moving by placing more paper strips, and varying the length of them to model different speeds. In each case, they count the number of infections. Students tabulate their results in the eight conditions: low and high population density, low and high population mobility, and whether or not to use face masks. Then, they write their conclusions and reflect on whether this is a computational model.

In order to make the detection of encounters easier, we can divide each paper strip into four parts (see Figure 9), and students find pairs of strips of paper that overlap in the same section. This is an easy way to solve simultaneous equations of discretized trajectories and it incorporates the fact that there is direct virus transmission only when two people are in the same place at the same time, and here, for simplicity, these models only consider direct transmission. For instance, in the scenario represented by Figure 9d, there are balls meeting at times 1 and 3. This kind of problem posing is encouraged in the APEC lesson study project [80] and Japanese lesson study [81] to develop mathematical thinking.

Figure 9a shows an orange ball at rest and the paper strip of the white ball that represents its trajectory when moving for a second. Figure 9b shows the trajectory of the same white ball but that it moves slower, covering half the path of the white ball of Figure 9a. In Figure 9c, two moving white balls are shown, and in Figure 9d, we increase population density to four moving white balls. In Figure 9e, larger balls representing people without masks are shown, and therefore, the corresponding paper strip path is wider.

This problem can be motivated by the real-world situation where some supermarkets have set times in the early morning that are exclusive for old people, in order to avoid the risk of virus transmission. This is because (1) old people walk slowly and do not run, whereas young people move faster and kids usually run; (2) in the early morning, there are less people. Different scenarios in Figure 9 can be interpreted as follows: (a) kid running, (b) adult walking, (c) and (d) many people moving fast, (e) without masks. The physical tapes represent the risk space for transmission.

#### 3.2.3. High School Students Who Learned Vectors and Probability

Here, the previous physical model and the idea of vector at the high school level is used. Students could try to estimate a mathematical expression for the probability of collision (contagion) depending on the population density, population mobility, and use of PPE. Let us consider two balls of radius r. The idea is to select a ball at rest (the black ball in Figure 10), then estimate the path of another ball (the red ball), and the probability to collide with the selected ball. If the red ball has speed v, then during the time interval from $t_{i}$ to $t_{f}$ the ball will move a distance $v\left(t_{f}-t_{i}\right)$ on the table leaving an imaginary trace of area $2rv\left(t_{f}-t_{i}\right)+\pi r^{2}$. We know there will be a collision if the center of the black ball is at a distance less or equal to r from this area. If we include all those points to the original area of movement, we obtain a total of $4rv\left(t_{f}-t_{i}\right)+\pi 4r^{2}$, and we call it the influence area of the red ball. Then, assuming that the black ball could be placed uniformly at random on the table, the probability of having a collision is proportional to this area. If we call A the area of the table, then the probability of collision is $\left(4rv\left(t_{f}-t_{i}\right)+\pi 4r^{2}\right)/A$.

If we have two balls at rest, one black and another blue, and keep the red one moving as shown in Figure 10b, then the area of influence is the same and assume the center of each of the balls at rest could be placed uniformly at random on the table then the probability of having a collision is twice as much as in the previous case.

Similarly, if we have n balls at rest, the influence area of influence of the moving ball is the same and then the probability of collision is n times the area, that is $n\left(4rv\left(t_{f}-t_{i}\right)+\pi 4r^{2}\right)/A$. The factor n/A corresponds to the population density. Thus, if we define d=n/A, we find that the probability of transmission is proportional to the product of the population density with the mobility and with the personal protective equipment (PPE) measured by the radius r.

What would happen if we have n balls moving at speed v?

Students can reflect on this simple model analyzing the possibility of overlapping traces and examining ways to overcome this effect. Moreover, they can consider what happens if the length of the time interval $\left(t_{f}-t_{i}\right)$ is too long.

These concepts can produce ideas for computer simulation and game programing. If we have real-world data, we can compare it with the simulation and fix the values of necessary parameters from real-world data.

In this exemplar, students have the opportunity to reflect on the viral transmission phenomenon for the computational model, analyze some factors that impact the transmission, build a concrete physical model for simplifying the variables, set the measurements varying the factors, build an odds model, compare predictions of their model with the results of the experiments, reflect on the limitations and strength of the model, and discuss ways to improve the model.

### 3.3. Exemplar 3: Machine Learning for Detecting Population at Risk

Machine learning (ML) is a branch of computer science that studies algorithms that learn to predict or classify. ML algorithms build mathematical or computational models based on a training data set. ML generates decision programs without being explicitly programmed to do so. After computing its accuracy in a ‘testing data set’, ML output can be used and seen as a kind of program.

Here, we illustrate an activity to simulate training a system to detect risk cases with COVID-19.

#### 3.3.1. Elementary School Students Who Are Able to Consider a Set with Conditions

Instead of using a computer, students can work with a classmate playing its role, to emulate training with a sample A of 30 people as the training data set. The ones that are sick have a pink painted mark on their foreheads, and the ones that are not sick do not have the pink mark (see Figure 11). The classmate has to figure out possible rules or patterns that can describe the sick patients. Then, on sample B, called the testing set, the classmate predicts who is infected and paints a pink mark on their foreheads. Then, the number of correct guesses is counted and the activity continues with a discussion considering the groupings using Venn diagrams and possible ways to improve the rule.

In this example with the training data set A (Figure 11), students can think inductively. Every person should be categorized with conditions such as infected (yes, no), age (old, adult, young, kid, baby), gender (male, female), hair color (white, black), hairstyle (none, short, long), beard (beard, mustache, none), glasses (yes, no), etc. Students should consider the conditions which may be related to infection and choose the necessary conditions to explain the relationship with infection. Then, classify the 30 people categorized into the groups which were defined as sets with conditions.

Following the bases for programing with CT, we ask students to express conditions in a computational form such as the rule:

IF gender = male, age = old, THEN infected

This rule is not perfect and students can then compute its accuracy. In our example with training set A, the accuracy of this rule is represented with the number of correct classifications and the number of misclassifications. For this particular rule, we have three correct and two incorrect classifications. In computational models, the accuracy is usually represented by subtraction or the ratio of these two quantities. Through this activity, by using training set A, students generate tentative rules for inductively finding an infected person. Then, students use test set B to apply the rule for prediction. The task sequence from A to B provides the metaphor for students how to produce an AI system with ML

#### 3.3.2. Middle School Students Who Learned Statistical Graphs

In middle school, students learn to analyze patients’ data and contribute to understand the risks factors. For these students, the activity is to analyze real-world COVID-19 data. The first 17 rows of a data set with COVID-19 deaths are shown in Table 1, a simplified version of data from [82]. This is the training data set. Before using ML, it is recommended that students learn what kinds of algorithms are used for this type of analysis. Here, the decision tree algorithm, a popular ML algorithm [83], is introduced.

The aim of this activity is to find characteristics of people that might predict whether COVID-19 will cause a fast death. The last column in Table 1 marks with ‘1′ patients with slow deaths, that is, those whose time from the first symptom to death higher than average, and zero those with fast deaths, which corresponds to time from the first symptom to death lower or equal than average. In this example, the average is 16 days. This variable can be used as the target for prediction.

Students can apply an algorithm that analyzes each variable independently (Gender, Age, 1st Symptom, Comorbidity, Surgery), and tries to measure for each variable how well it can classify patients into slow and fast death categories. For each variable, the algorithm positions the patients in a number line. For example, for the variable “Age”, students emulate the ML algorithm positioning every patient according to age and drawing with a circle those with slow death and with a cross those with fast death. Then, they will obtain Figure 12, a one-dimensional scatter graph.

From this scatter graph, students can try to detect a pattern. For instance, they could find that older patients die faster. Students then try to identify a critical age that best separates circles from crosses and generate a rule such as:

IF Age > critical age THEN fast death

In order to find a good critical age, for each possible value, students count the number of correctly classified patients and the number of incorrectly classified patients. For example, if they tentatively select 69 as the critical age, and then count those up to 69 years old that are classified as slower death and those older than 69 that are classified as faster death, then there will be a total of five misclassified cases. Students should try different critical ages and select the critical age with the least number of misclassifications. Once they select the critical age, they represent graphically the previous rule as a decision tree (Figure 13).

In this way, students emulate the process of a typical ML algorithm that was trained with data set A. As the next step, they are provided with patient data from local sources as a test dataset B; with that extra information and their rule, they can predict and compute the number of correct classifications and the number of misclassifications on test dataset B. They can compare the results on test data set B with those obtained on training dataset A.

#### 3.3.3. High School Students Who Learned Two-Dimensional Scatter Graph

Here, we extend the model to analyze two variables simultaneously by using two-dimensional scatter graphs and training data set A. The activity proposes that students analyze the other factors: gender, first symptom, comorbidity, and surgery. For each variable, they draw a number line (a line with just two positions for the binary variables). Are there any variables more influential to discriminate between fast and slow deaths? Similar to the previous activity, they emulate the training of an ML algorithm, such as decision trees [83,84,85]. In Figure 14, a two-dimensional scatter graph is shown for variables age and surgery.

In this example, 40% of patients with previous surgery have slow deaths, but 50% of those with no surgery have slow deaths. Students can also combine this with the critical age and write it as a rule. For example, they can detect the rule:

IF Age ≤ 69 AND Surgery = NA THEN slow death.

Then they can compute the number of correctly classified cases and the number of incorrectly classified cases and compare this result with those using only one variable.

By using spreadsheets, they can use the local patients’ data in order to build a big test set B with tens or hundreds of cases, make predictions for each patient in the test data set B, compute the number of correctly classified, and the number of misclassified cases.

Here, we illustrated one type of basic mechanism (algorithms) of ML, where students learn about training data set, test data set, features, induction, predictions, classifications, correct and incorrect classifications, critical thresholds, and graphic representations.

### 3.4. Lesson Study with Exemplar 2

To study the feasibility of these exemplars, taking advantage of the partial return to in-person classes, a 90 min session was held in several courses from two schools with a version of exemplars 2. The courses were for grades 4, 5, 6, 7, and 8 from two schools with a high degree of socioeconomic vulnerability according to the classification of the Chilean Ministry of Education. In total, 90 students participated.

A team of four teachers took turns teaching the sessions, while one taught the other three teachers observed the class and one of them videotaped the class. At the end of the session, it was automatically transcribed using speech recognition software. Then, based on the transcript and observations made, the teachers classified the stages of the session according to the COPUS protocol [86]. The teachers reviewed on the platform the automatic grading of the multiple-choice questions. The performance depends on the number of attempts to have a correct answer. Next, the teachers analyzed the written responses to the open questions, the comments the students wrote about their peer-written answers, and the verbal questions students posed during the class. Using that input, the teachers discussed their perception of the attitudes and interests of the students. The teachers then conducted a critical discussion. In addition, the teachers received our feedback. In this way, the teachers made adjustments to the lesson planning in order to improve the next session.

The session started with students logging in to the ConectaIdeas online platform [87,88], where they answered 31 questions.

They started answering 11 questions, 8 of which were open questions and 2 multiple choice questions with three options each one (questions 5 and 6). The questions were:

Question 1: What is COVID?

Question 2: Do you know someone who was infected?

Question 3: What factors do infections depend on?

Question 4: What causes infections to increase?

Question 5: Touch the stadium where you think there is a greater risk of transmission

Question 6: Touch the classroom with less risk of transmission

Question 7: To create a COVID game: What would the table or board look like?

Question 8: What would the balls or chips look like?

Question 9: What would the infections be like?

Question 10: What does the number of infected tokens or balls depend on?

Question 11: How would you represent people with masks?

The first two questions were designed to provide an opportunity for students to learn how to answer in the ConectaIdeas platform and get used to it. Questions 3 to 6 asked general information about infections and risks factors. Then, the teacher asked the students to imagine a board or table game about COVID-19. The objective was to know how they would model the infection mechanics. Then, questions 7 to 11 asked students about the game they have imagined.

Then, the teacher showed students the model with a toy pool table and some pool balls. The students then answered 17 multiple-choice questions. These are questions 12 to 28. The first three questions, questions 12 to 14, tested the basic understanding of the propagation mechanism. An example is shown in Figure 15.

The next two questions, questions 15 and 16, tested a deeper understanding that requires graphing the temporal evolution of infections. An example is shown in Figure 16.

Then the next four questions, questions 17 to 20, tested the notion of population density and its impact on the evolution of infections. Then, three more questions followed—questions 21 to 23. They tested the understanding of modeling and the use of facemasks. An example is shown in Figure 17.

Next, the last five questions, questions 24 to 28, tested the notion of population mobility and its impact on the evolution of infections. An example is shown in Figure 18.

Finally, the students answered three open questions, and the answers were peer reviewed with random assignment. The questions were the following:

Question 29: What problems or limitations do these models have to better represent reality?

Question 30: How would you include some people who die in the model?

Question 31: How would you include some recovered people in the model?

## 4. Results

A total of 90 students, 48 girls, and 42 boys participated in the experimental lesson study. They were 28 fourth graders, 7 fifth graders, 13 sixth graders, 19 seventh graders, and 23 eighth graders. The students belong to two schools with high socioeconomic vulnerability according to the indicators of the Ministry of Education. In one school, over 95% of the students are vulnerable and in the other school, 90% are vulnerable.

We conducted a lesson study in eight face-to-face lessons: two in fourth grade, two in seventh grade, two in eighth grade, one in fifth grade, and one in sixth grade. The first lesson was carried out in the eighth grade of the most vulnerable school. A set of four teachers participated. They took turns teaching the lesson, and after each lesson, they met to discuss it. They analyzed the class videos and adjusted for the next lesson. All classes were held in September 2021, once face-to-face classes began.

At the beginning of the lesson, before the model was presented, almost 80% of the students wrote that they knew someone that had COVID-19. Some of them were close relatives. Most students identified factors that are relevant for viral transmission such as social distance, agglomerations, use of facemasks, hands washing, and vaccines. However, none of them mentioned population mobility.

The answers to questions 7 to 11 regarding creating a game that models COVID-19 were very basic and based on superficial features. It seems that students have ideas to model the pandemic that are more complex than what they manage to express in written answers to open questions. For instance, answers to question 7 about the board to represent a COVID-19 game, showed great variability including:*“Square and indicating the different types of viruses”**“With dice and round chips”**“Cards with different types of transmission and other cards with protection to avoid infection”**“Like chess”**“It would be like Monopoly”*

It would be interesting to interact with them to let them expand their explanations and have a better understanding of their model designs with follow-up questions.

After the teacher presented the computational model, the average performance of the students in the 17 post-test multiple-choice questions was 78%. These are questions 12 to 28 and they test the understanding of the presented model. This score was automatically generated by the ConectaIdeas platform. It considers the number of attempts to obtain a correct answer. If we count the percentage of correct answers in the first attempt, the result is 45.3%. The 99% confidence interval is [40.26, 50.25]. Therefore, the result is well above chance, which is 25%. The multiple-choice question with the lowest-performing responses was the first question (Question 12). This is a question about the presented model that requests the use of the infection algorithm (as described in Section 3.1.1). The average performance was 66%. It is likely that the low performance may be due to being the first question about the model, where students were still learning how to answer this type of question in the platform. In the next two questions (13 and 14), the average performance was much better: 87% and 75%. The other low-performing questions were about population density. The question with the lowest performance had an average performance of 66%. Another question with low performance was a question about population mobility. The average performance was 71%. Three of the other mobility questions had an average performance of over 80%.

As expected, students from the most vulnerable school had a lower average performance than the other school. For example, in the fourth grade in the most vulnerable school, the percentage of correct answers in the first attempt was 36% while in the other school it was 46%. In the eighth grade, the percentage of correct answers on the first attempt in the most vulnerable school was 37% while in the least vulnerable school it was 56%. Additionally, within each school, the lowest performance was in fourth grade. Nonetheless, fourth-grade students from the most vulnerable school achieved an average performance of 72%. It is interesting to observe that the first multiple-choice question of the eighth-grade students of the most vulnerable school was the question where the lowest average performance was obtained—it had an average performance of 55%. This is the only question of all classes that had an average performance under the 60% barrier. This low performance was due to the fact that the first implementation was carried out in that class. Then performance improved. This improvement may be a product of the lesson study, where teachers were analyzing the lesson, making adjustments, and gaining experience.

In their answers to the final open questions that required reflection on the proposed model, its limitations, and also devising ways to overcome the limitations by introducing COVID-19 deaths and recoveries, the students gave a wide range of answers. Some of them reveal an understanding of modeling and how to improve it. For example, some responses to limitations were:*“It is difficult to represent reality”**“It’s hard to represent the truth”**“Everyone would have to touch each other”**“The limitation of my model is that the infection is minimal”**“I think they are good since they are easy to understand, but the mobility model should move in different directions, but apart from that I find everything is OK”*

Some students also reflected on the effect of mobility on transmission. It is interesting that some included this factor since it was not mentioned before the teacher presented the model.

Regarding representing the deaths, many students devised ways to distinguish these balls by color or other physical attributes. Some students suggested removing those balls from the table. Regarding the recovered ones, many devised to change their color. Others proposed going back to the original color. Finally, everyone commented on the responses of the peer who was randomly assigned to them. The majority supported what was expressed by the classmate, but there were some who criticized what was proposed. For example, a comment was the following:


*“Sounds good to me, but I think it’s better if it wasn’t so simple. Maybe add more things, but only proposing an angel (for the dead) is poor. If I want a game that is something better. I’d put something else on it.”*


In summary, students were able to propose ideas to extend the model to include more features such as deaths or recovered cases. Although their proposals might not be the most ‘efficient’, they correctly represent what is required. As expected, concrete thinking was more common among fourth-grade students. They tended to identify modeling features as limitations, e.g., “*people are not balls*”, “*it lacks images of people with or without facemasks*”, representing deaths with skulls, etc. Older students wrote longer answers, identified deeper limitations, and devised better ways to overcome the limitations.

## 5. Discussion

In order to answer our research question, we have designed three exemplars of CT learning activities and we used the Japanese Lesson Study methodology to implement the computational modeling exemplar. A team of four elementary and middle school teachers taught a lesson based on the exemplar of computational thinking. In total, each teacher did 2 to 3 sessions of the same lesson with courses from fourth to eighth grade. We conducted the lesson study in two schools of high socioeconomic vulnerability.

Although we have carried out a case study only with one exemplar, the empirical results are promising and open the opportunity for further research on the topic. Moreover, the main contribution of the paper is well beyond the empirical results, since we have shown that is possible to design learning activities that integrate CT with real-world problems, particularly, the COVID-19 pandemic.

We observed that students understood at some reasonable level the model proposed. They understood the different components, the way to include in the model the use of masks and social distance, and the effect of population density and mobility. Additionally, they understood the concept of iteration and how the dynamics of viral transmission are represented in the model.

Given the positive results in these schools, we could be able to demonstrate that this innovative and challenging proposal was feasible. We expect this to happen in schools that are less vulnerable as well since they systematically perform better according to standardized national tests. Another reason for the interest in vulnerable schools is that this socioeconomic sector has been the most affected by the pandemic [89]. Therefore, it is critical to adapt mathematics and science literacies and critical thinking to improve citizens’ behavior during the pandemic. The result of the implementation of the computational modeling exemplar indicates that students can understand the computational model: they managed to connect it with the real problem, understand some of the limitations of the model, and were able to reflect effectively on how to extend it to overcome some of its limitations. Although each class had a very short experience, only 90 min long, this preliminary study allows us to estimate that a sequence of more lessons can be successfully implemented in schools. On the other hand, the team of teachers that implemented the exemplar and conducted the Lesson Study, gradually gained confidence, analyzed the lessons together, and thus improved the lesson. Additionally, the approach we have followed to develop computational thinking focuses on the fundamentals of CT and is conducted long before students experience their first programming language as some researchers recommend [90]. It also promotes integration with the mathematics and science curricula, which is a growing need in engineering and science practice, but it is still a major unresolved educational challenge in the school system [91].

There are still several aspects that must be investigated in the near future. First, it is necessary to implement the other two exemplars proposed in this paper: the exemplar for algorithmic thinking and the one for teaching machine learning. Second, we need to conduct the study of classes in these two exemplars, analyze the impact on students´ learning, and study the adaptation process of teachers to achieve a more effective lesson. Third, it is important to test these exemplars in a larger number and more heterogeneous sample of schools. Fourth, it is also crucial to design randomized controlled trials (RCT) [92] that can be compared with randomly selected controls in order to accurately determine the effect size and the statistical robustness of the findings. Fifth, it is necessary to design a longer sequence of sessions that allows for a more profound impact on students and a deeper connection with the mathematics and science curriculum. Indeed, the long-term impact of CT education is not yet fully understood, therefore, a natural follow-up study would be to design and apply an assessment to gain some insight into long-term learning outcomes. Sixth, it is also necessary to design ways to measure and estimate the long-term impact on student behaviors, values, and attitudes on a larger scale, without teachers having to read all written answers, and actively discuss them with the students. Seventh, it is ideal to be able to study to what extent these exemplars help to achieve the UNESCO SGD and how it compares to other activities.

## 6. Conclusions

In summary, we can conclude that it is feasible to introduce into the computational thinking curriculum a sequence of exemplar lessons that address critical public health strategies to improve decision making in pandemics. It is possible to successfully introduce the exemplars for the three pillars as defined by APEC InMside and at all grade levels and in schools of high vulnerability. Students were able to understand a computational model of the pandemic that captured some of its key features. They were also able to analyze the model and connect it with the corresponding graphs of the progression of the contagion phenomena. Additionally, they were aware of some of the limitations of the model. Students reflected on these limitations and devised ways to overcome them. They proposed some strategies and representations to include deaths and recoveries. Moreover, they could also understand the factors that influence pandemics such as population density, population mobility, facemask use, social distance, and ways to control them. Undoubtedly, the lack of understanding of key aspects of these factors has affected virus transmission in the current COVID-19 pandemic. Hence, the exemplars of lessons introduced in this work are a promising example of STEM integration that helps reflect and tackle real-world and challenging public health problems of great impact for students and their families.

## Figures and Tables

**Figure 1 ijerph-18-12520-f001:**
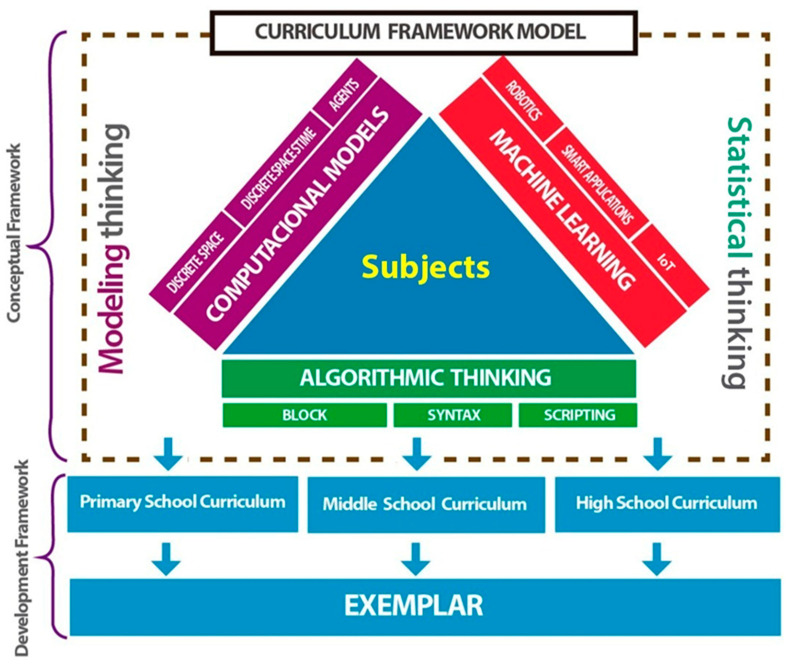
Curriculum framework model.

**Figure 2 ijerph-18-12520-f002:**
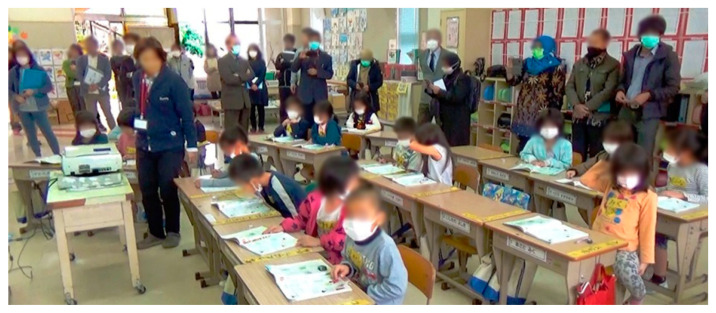
A lesson study: Participants and students wear masks to avoid COVID-19 infection.

**Figure 3 ijerph-18-12520-f003:**
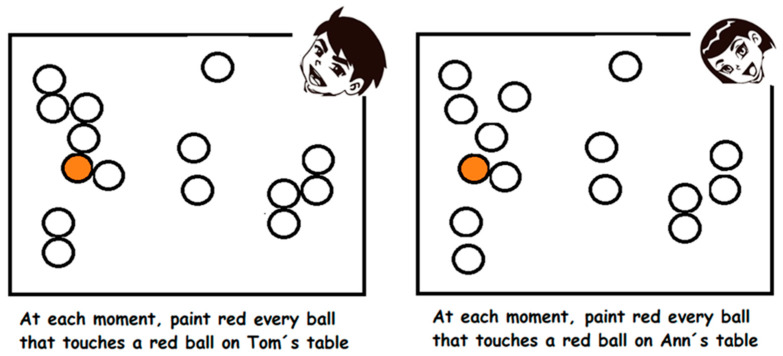
A coloring activity with a recursive painting instruction, where contagion in a ball is simulated by painting it red, as in (NHK, 2020) experiment.

**Figure 4 ijerph-18-12520-f004:**
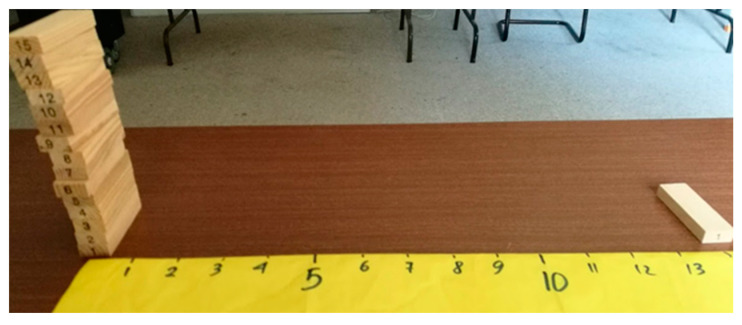
At the left end of the ruler 15 blocks representing 15 PCR-positive people, and 14 days later one of these people dies (shown at the right end).

**Figure 5 ijerph-18-12520-f005:**
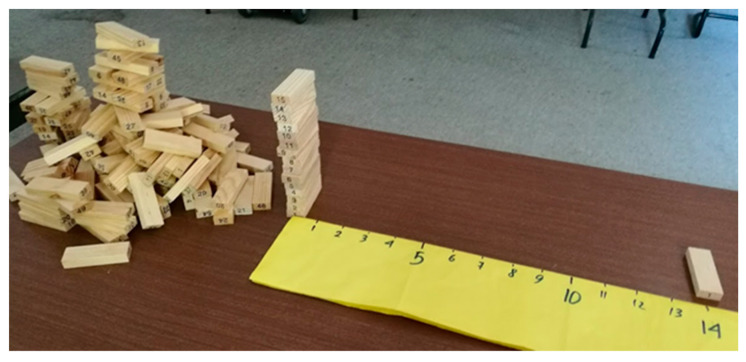
At the left of the image, a pile of blocks representing the unknown number of infected people on day 0 for each person that dies on 14th day.

**Figure 6 ijerph-18-12520-f006:**
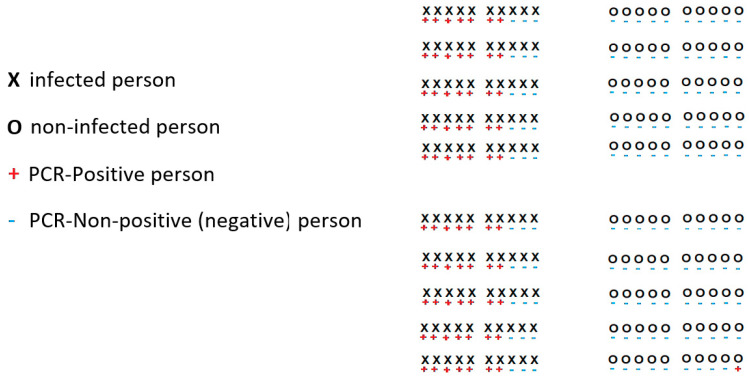
One hundred infected persons and one hundred non-infected persons. Seventy of the infected persons are PCR positive, and 99 of the non-infected persons are PCR negative. Thus, sensitivity is 70% and specificity is 99%.

**Figure 7 ijerph-18-12520-f007:**
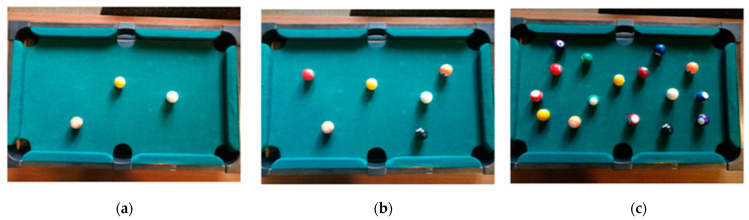
Different number of balls on the pool table to represent (**a**) low, (**b**) medium and (**c**) high population densities.

**Figure 8 ijerph-18-12520-f008:**
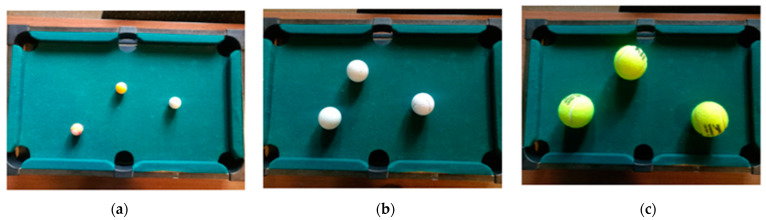
PPE and social distance protocol represented by the size of the balls. With face masks of high quality as shown in images (**a**), with ordinary masks in (**b**), with no mask in (**c**). In (**a**) people must be very close in order to have a higher probability of transmission whereas in (**c**) it is enough to be a couple of meters from each other to transmit the virus.

**Figure 9 ijerph-18-12520-f009:**
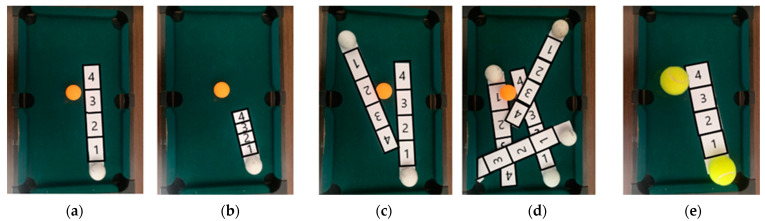
Balls on a pool table modeling the probability of collision (virus transmission) under the different conditions of population density, population mobility, and use or not of face masks. They represent: (**a**) kid running, (**b**) adult walking, (**c**) and (**d**) many people moving fast, (**e**) without masks.

**Figure 10 ijerph-18-12520-f010:**
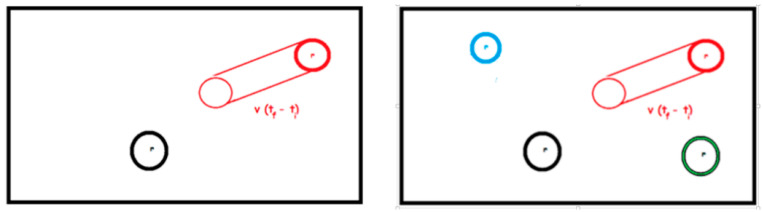
(**a**) Red ball moving at velocity v leaving a trace on the floor of the pool table. (**b**) Three balls at rest and one ball moving.

**Figure 11 ijerph-18-12520-f011:**
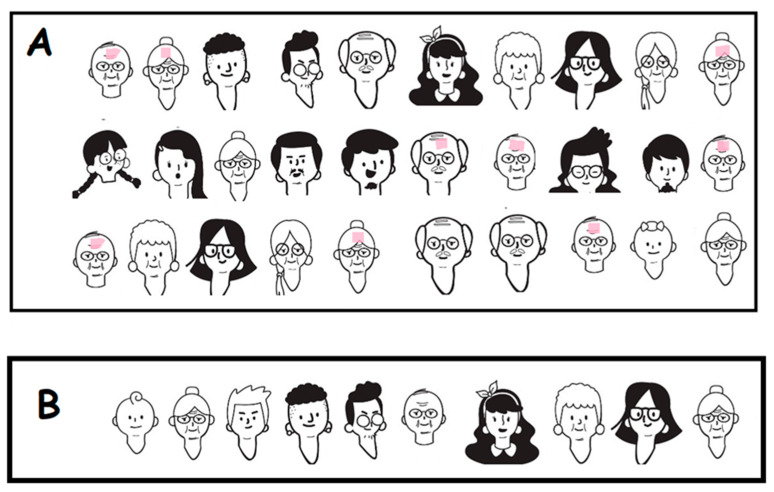
Training set (**A**) and test set (**B**), with patients. The ones in the training set (**A**) with pink mark on their foreheads are ill.

**Figure 12 ijerph-18-12520-f012:**
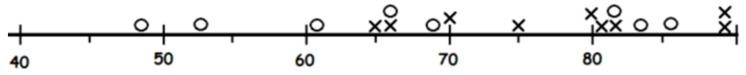
One-dimensional scatter graph with first symptom to death below and above average categories.

**Figure 13 ijerph-18-12520-f013:**
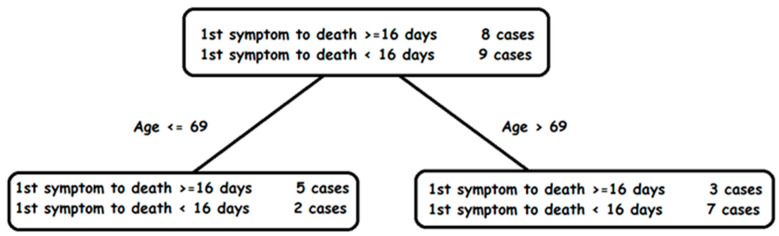
Decision tree obtained from the scatter graph.

**Figure 14 ijerph-18-12520-f014:**
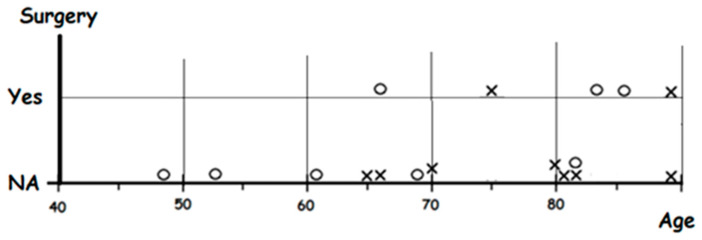
Two-dimensional scatter graph with variables surgery and age.

**Figure 15 ijerph-18-12520-f015:**
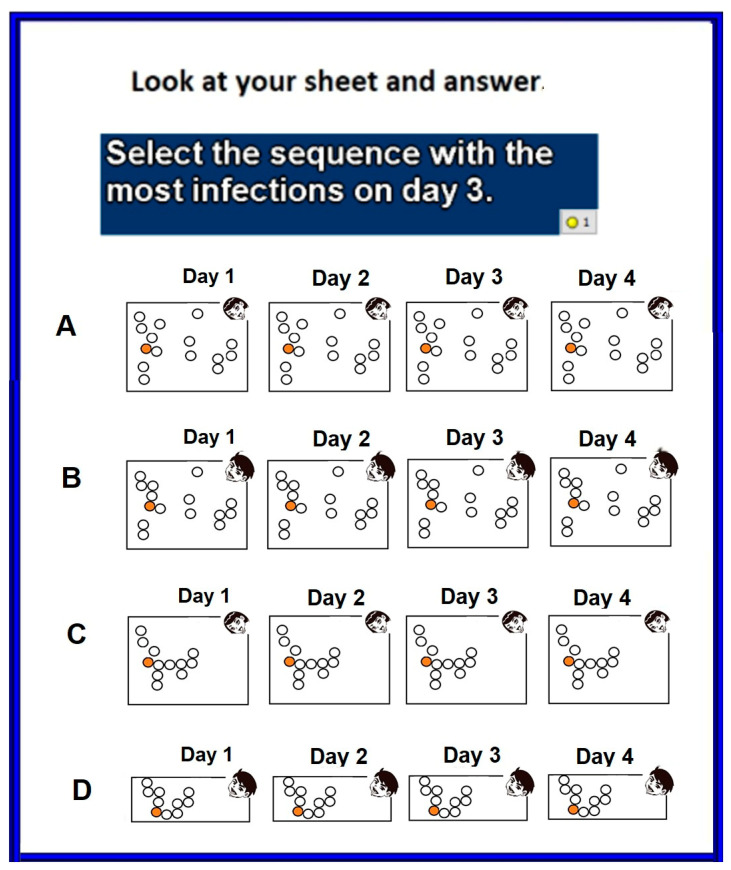
Example of a question that tested whether students understand a model of viral transmission across different days as a sequence of pool balls touching each other on a pool table.

**Figure 16 ijerph-18-12520-f016:**
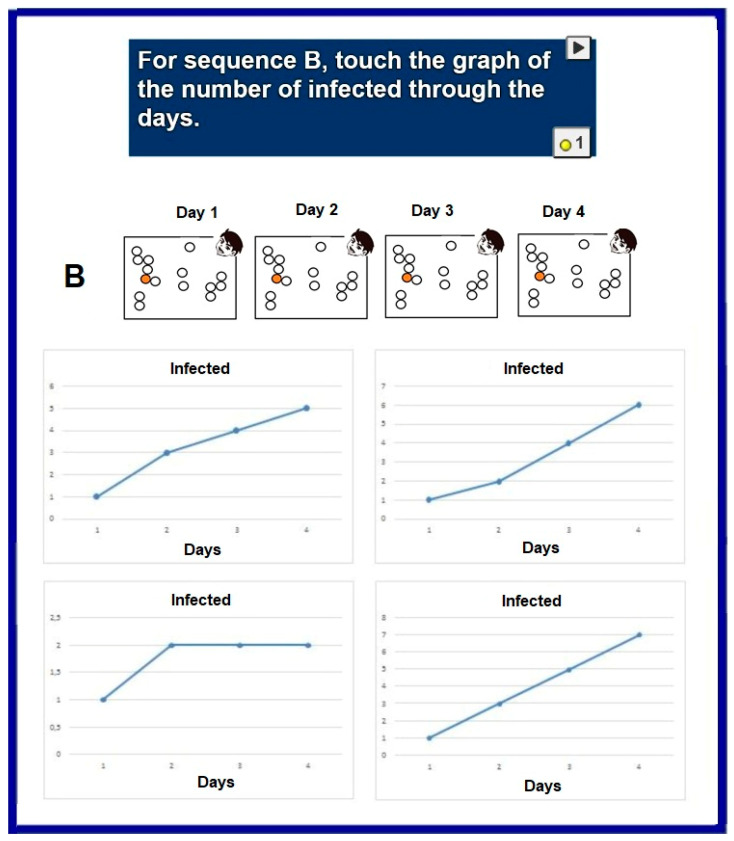
Example of a question that tested whether students understand the pool table model representing transmission across different days and are able to plot it.

**Figure 17 ijerph-18-12520-f017:**
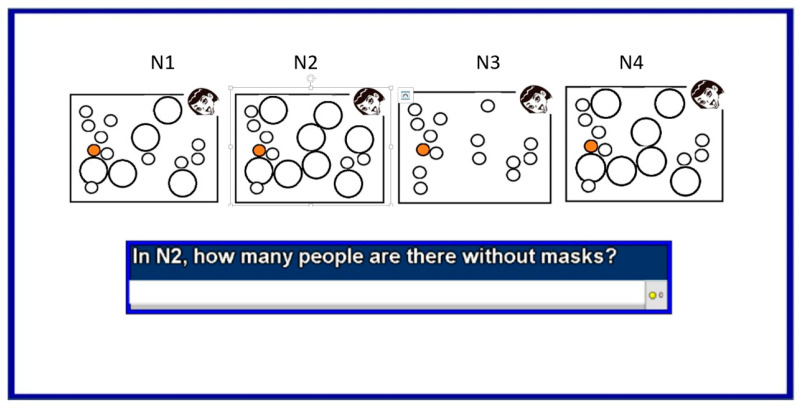
Example of a question that tested whether students understand the pool table model representing transmission across different days and are able to plot it.

**Figure 18 ijerph-18-12520-f018:**
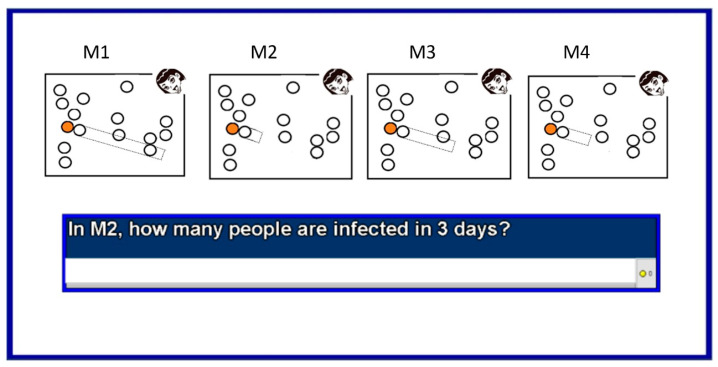
Example of a question that tested whether students understand an extension of the pool table model that includes population mobility.

**Table 1 ijerph-18-12520-t001:** The first 17 rows of a data set with COVID-19 deaths, adapted from [82].

Case	Gender	Age	1st Symptom	Comorbidity	Surgery	1st Symptom to Death (days)	1st Symptom to Death above Average?
1	M	61	Fever	yes	NA	20	1
2	M	69	Fever	yes	NA	16	1
3	M	89	No Fever	yes	NA	10	0
4	M	89	No Fever	yes	Yes	6	0
5	M	66	Fever	yes	Yes	10	0
6	M	75	Fever	yes	Yes	14	0
7	F	48	Fever	yes	NA	41	1
8	M	82	No Fever	NA	NA	12	0
9	M	66	No Fever	NA	NA	30	1
10	M	81	Fever	NA	NA	11	0
11	F	82	Fever	yes	NA	19	1
12	M	65	No Fever	NA	NA	13	0
13	F	80	Fever	yes	NA	11	0
14	M	53	Fever	NA	NA	20	1
15	M	86	No Fever	yes	Yes	19	1
16	F	70	Fever	NA	NA	8	0
17	M	84	Fever	yes	Yes	16	1

## Data Availability

Not applicable.

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
