# Peer review of "Developing Computational Thinking Teaching Strategies to Model Pandemics and Containment Measures"

_ijerph, 2021, doi:10.3390/ijerph182312520_

Round 1

Reviewer 1 Report

I liked reading your paper as it demonstrates the importance of CT education with a specific and very timely problem. Though I'd have liked clarification on the exact definition of CT you are subscribing to, as explained in APEC InMside, I appreciated the practicability and clarity of your report. Specifically the escalation of knowledge between different grades was well executed.
As a non-native speaker I stumbled over your usage of "exemplar" a lot during the reviewing process and I do assume you often mean "example".

I will end my review with a suggestion for future research: The assessment and especially the long term impact of CT education is not yet fully understood, with this project you could envision a follow up assessment and gain some insight in long term learning outcomes.

Overall a very clear research project, well worth a read. Thank you.

Author Response

Dear reviewer.

Thank you very much for your time and dedication to reviewing our paper. We appreciate very much that you find the paper provides valuable information.

Reviewer 2 Report

This manuscript presents an interesting approach to introduce computational thinking into STEM curricula with a focus on the current pandemic. While the ideas presented are interesting I believe the manuscript needs substantial revision as it presents a lot of details regarding the exemplars but very little on the study meant to address the research question. Furthermore the manuscript lacks conciseness. Below I present the issues in further detail. 

Title - the title is very broad and not specific on the work conducted.

Introduction -  the introduction is long, providing details that do not seem relevant while not providing information on others, relevant to the public health audience. In particular:

  1. Public health readers might not be familiar with the concept of "exemplar" in education. I suggest to carify
  2. Lines 54-59 lack references
  3. Lines 81-95. This paragraph presents several ideas not covered in the paper and also lacks references. The authors also overemphasize the role of models as some of the information regarding disease control comes from other existing knowledge. I would also argue the issue is not the "effectiveness of the models" (last sentence) but the effectiveness of the measures. This goes way beyond the understanding of the models. Please comment. 
  4. Lines 96-101. This paragraph partially addresses some of the concerns raised in the previous one. It seems more relevant to combine these two. 
  5. Lines 109-118. I find it difficult to see the relevance of this paragraph in this manuscript. 
  6. The final paragraphs of the introduction focus on the "intervention" conducted. I would find it more appropriate to combine this on the description of the approach considered.

Methods

  1. This section is exceptionally long and does not follow the usually approach for a manuscript in public health. I find the exemplars description very interesting but it is really easy to get lost within the work conducted. As far as I understood the lesson study did not consider all the exemplars - is this correct? If so, I believe that only the exemplars considered should be explained in detail. Perhaps the remaining could be moved to supplementary material? 
  2. It is unclear how the data was analysed (line 708). Please clarify.
  3. I cannot see what is represented in the plots of figure 16 

Results

  1. Results reporting is quite poor and limited. It is unclear what the questions refer to and how the answers were analysed. 
  2. The authors do not provide enough information to support their statements. For example how do the authors compared students between schools?
  3. Lines 810-814 - this seems more discussion/conclusions than results. 

Discussion

The discussion is extremely limited, with a lot of repetition and not covering essential points:

  1. The 3 first paragraphs are quite a long summary of what was done. Consider to review. 
  2. Lines 865-867 - How exactly the authors assume this? Can they say that the schools are representative of all schools? How was this ensured? This statement is more puzzling when one reads lines 885-86 which contradicts the lines abovementioned.
  3.  Lines 890-891 - Wasn't this what you have done? Are you suggesting that your study did not address its own objective? 
  4. Discussion of the results in comparison with existing literature in light of the objectives is lacking

Conclusion - unnecessarily long and providing more of a summary than the actual conclusion. 

Other points:

- The authors stated they did not require ethics approval. However they provide a figure with the students in the class. Did they collected appropriate consent? 

- Line 531 - this statement only applies when we consider direct transmission. Please revise. 

Author Response

Dear reviewer.

Thank you very much for your time and dedication to doing a thorough revision of the manuscript.

Reviewer 3 Report

This research is appropriately organized and provides valuable information. Authors should address all of the following concerns carefully.

  • Abstract: What are STEM and APEC InMside? The abstract should be comprehensive and independent.
  • Introduction Section: We suggest that the authors add the main contributions as bullet points at the end of the introduction.
  • There are some acronyms that you should define before using them such as AI, IoT … etc.
  • Equations: Symbols in equations should be italicized like the equation 1, 2 … etc.
  • Discussion Section: We suggest removing the first paragraph in the discussion section as it is unnecessary.
  • Conclusion Section: It is long and prefers to reduce the number of words.
  • Figures and Tables: Figures 10, 13, 15, 16 need to be enlarged to be clearer. Figure 11 is shown before it is used in the text. What is the use of the colors in Figure 1? Table 1 should be rewritten to be clear and accurate.
  • Paraphrase text: There are many sentences taken from previous papers. Authors must paraphrase all sentences and phrases taken from the previous papers even if research for the same authors. The authors must avoid taking whole sentences from the original papers. They are obligated to paraphrase during the revision of the entire paper.
  • “a critical cognitive disposition that students need to adopt and 40 use efficiently to solve the challenges of the establishment of digital society” (page1), “Algorithmic thinking emerges when someone observes repeated patterns in problems and then generalizes a set of rules for dealing with such situations (so that one does not need to think this through each time the problem occurs). In effect, one creates (formally or informally) an algorithm or rational procedure. This sort of thinking is often used in problem solving and in computer programming.” (page4),

.

.

.

Etc.

  • English Writing: This paper requires moderate proofreading to address some typographical and grammatical issues and to make English writing readable and understandable. This revision requires careful proofreading to improve the English writing.
  • References list: References are recent, adequate and relevant to the research topic. However, authors should remove all mistakes in the reference list. For example, some references do not contain enough information such as the references [4], [5], [6], [7], [18], [51] … etc. References should follow the MDPI-IJERPH Some search names in the reference list begin an uppercase letter for each word (such as [15], [29] ... etc.) and others use only an uppercase letter in the first word (such as [16], [31] … etc.), authors should standardize style. The reference list needs minor improvement.

Author Response

(The authors gave the same response as above.)
